# Does reactogenicity after a second injection of the BNT162b2 vaccine predict spike IgG antibody levels in healthy Japanese subjects?

**Masaaki Takeuchi**[1]*, **Yukie Higa**[1], **Akina Esaki**[1], **Yosuke Nabeshima**[2], **Akemi Nakazono**[1]

**1** Department of Laboratory and Transfusion Medicine, Hospital of University of Occupational and Environmental Health, School of Medicine, Kitakyushu, Japan, **2** Second Department of Internal Medicine, University of Occupational and Environmental Health, School of Medicine, Kitakyushu, Japan

* takeuchi@med.uoeh-u.ac.jp

**Data Availability Statement:** All relevant data are within the manuscript and its Supporting information files.

## Abstract

### Background

Adverse reactions are more common after the second injection of messenger RNA vaccines such as Pfizer/BioNTech's BNT162b2. We hypothesized that the degree and severity of reactogenicity after the second injection reflects the magnitude of antibody production against the SARS CoV-2 virus spike protein (spike IgG).

### Methods and results

Blood samples were obtained from 67 Japanese healthcare workers three weeks after the first injection and two weeks after the second injection of the BNT162b2 vaccine to measure spike IgG levels. Using questionnaires, we calculated an adverse event (AE) score (0–11) for each participant. The geometric mean of spike IgG titers increased from 1,047 antibody units (AU/mL) (95% confidence interval (95% CI): 855–1282 AU/mL) after the first injection to 17,378 AU/mL (95% CI: 14,622–20,663 AU/mL) after the second injection. The median AE score increased from 2 to 5. Spike IgG levels after the second injection were negatively correlated with age and positively correlated with spike IgG after the first injection. AE scores after the second injection were not significantly associated with log-transformed spike IgG after the second injection, when adjusted for age, sex, AE score after the first injection, and log-transformed spike IgG after the first injection.

### Conclusions

Although the sample size was relatively small, reactogenicity after the second injection may not accurately reflect antibody production.

**Funding:** The authors received no specific funding for this work.

**Competing interests:** MT received a research grant from Abbott. Other authors declare that they have no conflicts of interest. This does not alter our adherence to PLOS ONE policies on sharing data and materials.

## Introduction

Effective and appropriate coronavirus disease 2019 (Covid-19) vaccination is the most promising strategy for controlling the spread of SARS-CoV2 infection on a worldwide basis [1–3]. Currently, nucleoside-modified messenger RNA (mRNA) vaccines encoding SARS-CoV2 full-length spike and adenoviral vector vaccines have been used mainly in Western countries. Vaccination levels in Canada, the United States, England, and Israel are now reaching 55%–69%, dramatically reducing the infection rate [4–7]. In Japan, the number of Covid-19 infections is increasing, but the vaccination rate is still low (26%) due to a limited supply of vaccines and a slow throughput for vaccine injection.

Local and systemic reactogenicity after the second injection of mRNA vaccines is more common than after the first injection [1, 2, 8]. The second injection has a booster effect that produces substantial antibody titers against SARS-CoV2 spike antigen. Thus, we hypothesized that the degree of reactogenicity following the second injection could be an indicator of the level of SARS-CoV2 spike antibody production. Accordingly, the aim of this study was to investigate whether local and systemic reactogenicity after the second injection of an mRNA vaccine reflects subsequent SARS-CoV2 spike antibody levels in Japanese healthcare workers.

## Materials and methods

### Study participants

This was a prospective, longitudinal, observational study in a single center. The study was approved by the institutional review board of the University of Occupational and Environmental Health, School of Medicine (approval number: UOEHCRB21-023). During early 2021, the only vaccine available in Japan was the BNT162b2 mRNA Covid-19 vaccine (Pfizer/BioNTech). The Japanese government started to distribute this vaccine to healthcare workers in February 2021, and our university hospital received the vaccine in the middle of March 2021. The hospital chairman decided to administer the first dose of BNT162b2 mRNA Covid-19 vaccine (30 μg per dose injected into the deltoid muscle) to hospital employees during the fourth week of March and the second dose during the third week of April (April 12 to April 16). Since we received ethical approval for the study on April 12 the time available to acquire informed consent and to conduct the study was quite short (range: 1 to 5 days), we advertised for the study participation to subjects working in four departments (Department of Laboratory and Transfusion Medicine, Department of Pathology, Department of Pharmacy, Second Department of Internal Medicine) in the hospital. Study participation was voluntary. Participant recruitment was started on April 12 and ended on April 16. Written informed consent was obtained from all personnel who agreed to participate.

### Antibody test

Blood samples were obtained from participants before the second dose of the BNT162b2 mRNA vaccination (median: 20 days [interquartile range (IQR): 20 to 21 days] after the first dose) and 2 weeks after the second dose (median: 13 days [IQR: 11 to 14 days] after the second dose). A SARS-CoV2 IgG assay was performed with chemiluminescent immunoanalysis of microparticles used for quantitative detection of IgG antibodies against the spike protein of the SARS CoV-2 virus (spike IgG) employing the Architect system (Abbott Diagnostics) with a cut-off <50 antibody units/mL (AU/mL) in both blood samples. To exclude the possibility of previous Covid-19 infection, we also measured IgG antibodies against the nucleocapsid protein of Covid-19 (Abbott Diagnostics) using the second blood sample. The negative cut-off value for anti-nucleocapsid protein IgG was < 1.4 AU/mL.

## Reactogenicity

All participants were asked to respond to questionnaires after both the first and second blood samples regarding local and systemic adverse effects to the vaccine injections. Questionnaires included a previous history of allergic reactions, severe adverse effects such as anaphylactic reactions, local reactogenicity such as pain at the injection site, swelling, lymph node swelling, and systemic reactogenicity, including fatigue, headache, myalgia, arthralgia, chills, and fever (body temperature $\geq 38$˚C). To estimate reactogenicity, we calculated an adverse effect score (AE score). For each symptom present, whether local and systemic, we assigned a score of 1. The exception was fever. A score of 1 was given if body temperature increased to $\geq 38.0$˚C but $< 38.4$˚C. A score of 2 corresponded to a body temperature of $\geq 38.4$˚C to $< 38.9$˚C. Development of a body temperature $\geq 38.9$˚C was scored as 3 [2, 3]. We summed all scores and calculated AE scores after the first and second injection (range: 0 to 11).

The primary outcome was to determine whether the AE score after the second injection was associated with spike IgG levels two weeks after the second injection.

## Statistical analysis

Levels of spike IgG were expressed as the median, IQR, and the geometric mean and 95% confidence intervals calculated using Student's t test distribution on log-transformed data. AE score was shown as the median and IQR. Categorical data are presented by number and percentage. Given the small number of participants, we did not perform formal statistical comparisons between groups. Spearman's correlation analysis was used to compare spike IgG levels after the first and second injections and between spike IgG after the second injection and age. Correlation analysis was further performed on subgroups stratified by gender. To predict the level of log-transformed spike IgG two weeks after the second injection, multivariate linear regression analysis was performed, using age, sex, log-transformed spike IgG three weeks after the first injection, and AE scores after the first and the second injection as covariates. A two-sided p-value $< 0.05$ was considered significant. All statistical analyses were conducted using R software version 4.0.4 (R foundation for Statistical Computing, Vienna).

## Results

Among 109 healthcare workers who were working in any of the four Departments, 69 subjects agreed to participate the study. However, two participants did not come to receive a second blood sample examination within the specified time window; thus, we excluded them from the analysis. The final study population comprised 67 participants (age 43 years [IQR: 32–51 years], 21 males). Six subjects reported previous histories of allergic reactions.

## Reactogenicity

No one developed a severe adverse reaction after the vaccinations. Fig 1 presents local and systemic reactogenicity after the first and second injections. Except for myalgia, the prevalence of all other reactogenicities increased from the first injection to the second injection, resulting in a higher AE score after the second injection (median: 5, IQR: 2–6) than after the first injection (median: 2, IQR: 1–3). Since previous studies showed higher reactogenicities in younger subjects [2, 9], we divided subjects into two groups stratified by age and compared reactogenicities after the second injection. Reactogenicity was higher among subjects $< 50$ years than among those $\geq 50$ years (Table 1). Thus, AE scores after the second injection were higher in subjects who were $< 50$ years (median: 5, IQR: 4–6) compared with subjects $\geq 50$ years (median: 2, IQR: 1–3).

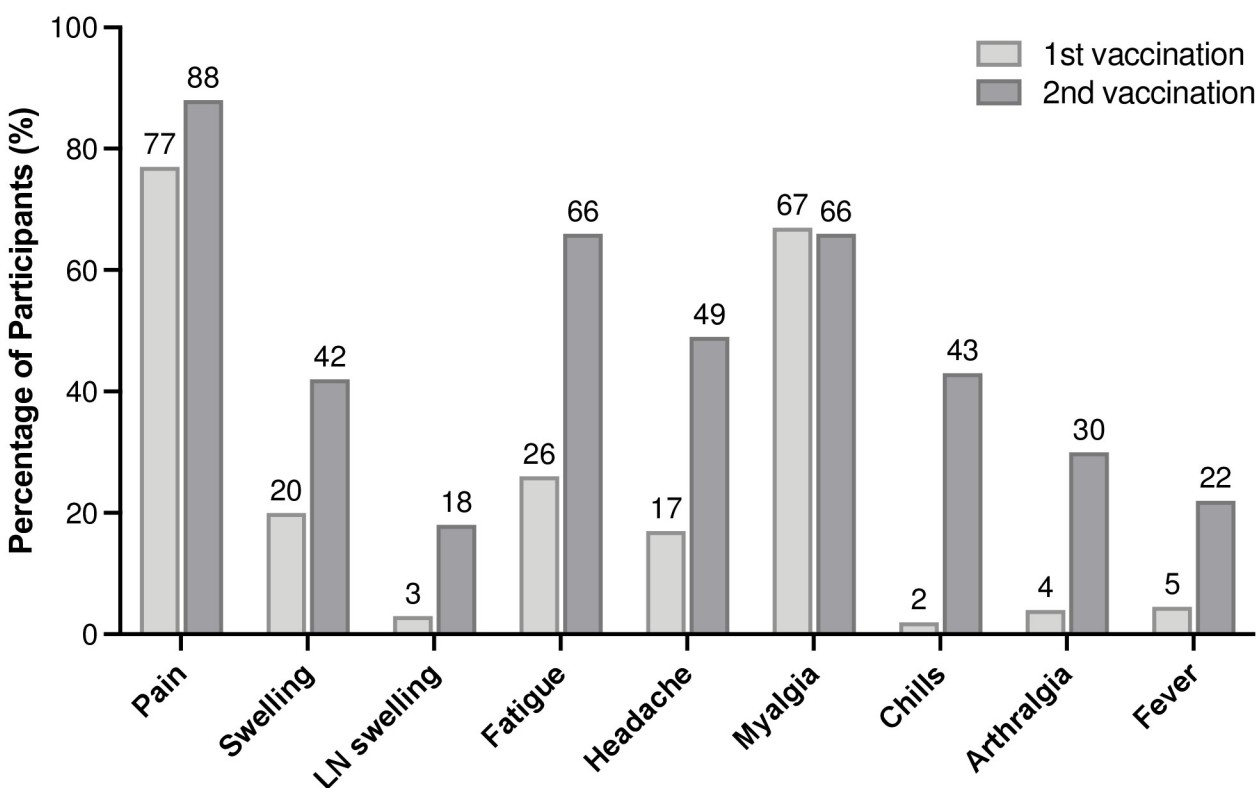

**Fig 1. Local and systemic reactions after the first and second injections of the Pfizer/BioNTech BNT162b2 vaccine among all subjects.** Fever was defined as a body temperature ≥ 38˚C after a vaccine injection.

## Antibodies

None of the subjects were positive for anti-nucleocapsid IgG (median: 0.04 AU/mL, IQR: 0.02–0.07 AU/mL). All subjects were positive for spike IgG (> 50 AU/mL), for which the median, IQR, geometric mean and its 95% confidence interval after the first injection were 1,065 AU/mL, 741–1,693 AU/mL, 1,047 AU/mL, and 855–1,282 AU/mL, respectively. After the second injection, corresponding values increased to 17,259 AU/mL, 12,088–28,999 AU/mL, 17,378 AU/mL, and 14,622–20,663 AU/mL, respectively, a 16-fold increase of spike IgG from the first to the second injection. There was a positive correlation between spike IgG levels after the first injection and after the second injection (r = 0.62, p < 0.001, Fig 2A). There was also a negative correlation between age and spike IgG levels after the second injection (r = -0.37, p = 0.002, Fig 2B).

## Linear regression analysis

Univariate linear regression analysis revealed that age, log-transformed spike IgG after the first injection, and AE score after the second injection were associated with log-transformed spike IgG after the second injection. However, sex and AE score after the first injection was not associated with log-transformed spike IgG after the second injection (Table 2).

Multivariate linear regression analysis revealed that age and log-transformed spike IgG after the first injection were significantly associated with log-transformed spike IgG after the second injection. However, sex, AE score after the first injection, and AE score after the second

**Table 1. Comparisons of adverse reaction and spike IgG titers after the second injection of BNT162b2 mRNA vaccine among subjects ≤ 50 years and those > 50 years.**

| Variable | Age ≤ 50 years (n = 46) | Age > 50 years (n = 21) |
|---|---|---|
| Male gender | 14 (30%) | 7 (33%) |
| Spike IgG (AU/mL) | 20,070 (IQR: 13529–33898) | 14,766 (10131–19693) |
| Pain at injection site | 42 (91%) | 17 (81%) |
| Swelling | 23 (50%) | 5 (24%) |
| LN swelling | 10 (22%) | 2 (9.5%) |
| Fatigue | 37 (80%) | 7 (33%) |
| Headache | 28 (61%) | 5 (24%) |
| Myalgia | 32 (70%) | 12 (57%) |
| Chills | 26 (57%) | 3 (14%) |
| Arthralgia | 18 (39%) | 2 (9.5%) |
| fever | 13 (28%) | 2 (9.5%) |
| Degree of fever | | |
| < 38.0˚C | 33 (72%) | 19 (90%) |
| ≥ 38.0 but < 38.4˚C | 7 (15%) | 2 (10%) |
| ≥ 38.4 but < 38.9˚C | 3 (6.5%) | 0 (0%) |
| ≥ 38.9˚C | 3 (6.5%) | 0 (0%) |
| AE score | 5.0 (4.0–6.0) | 2.0 (1.0–3.0) |

Continuous values are expressed as median and interquartile range (IQR).

AE, adverse effect.

injection were not significantly associated with log-transformed spike IgG after the second injection (Table 3). After removing those three variables from the model, the final model yielded an F-statistic of 52.4, a residual standard error of 0.29, and an adjusted $R^2$ of 0.61.

## Discussion

The main findings of this study can be summarized as follows: (1) the prevalence of local and systemic reactogenicity increased approximately 2.5 times after the second injection, compared to the first injection, and its severity was more significant in younger people; (2) spike IgG titers became positive in all participants after the first injection, and those titers increased 16-fold after the second injection; (3) spike IgG titers after the second injection were negatively correlated with age, and positively correlated with spike IgG after the first injection; (4) multivariate linear regression analysis revealed that the degree of reactogenicity after the second injection was not significantly associated with spike IgG levels after the second injection after adjusting for age, gender, AE score after the first injection, and spike IgG levels after the first injection.

None of the subjects in this study showed positive results for nucleocapsid IgG; thus, we believe that all participants were Covid-19 naïve subjects [10]. Although the degree of adverse effects after the first injection was similar to those in a previous publication from western countries [2, 3, 11], the prevalence of reactogenicity after the second injection was higher. Although proposed causes may be related to racial differences and geographic differences of non-COVID-19 corona virus exposure, we should consider a fixed dosing regimen as a potential cause of different prevalence of adverse reactions. Dose-dependent increases in adverse effects have been observed with mRNA vaccines [1–3] and the current recommended dose of

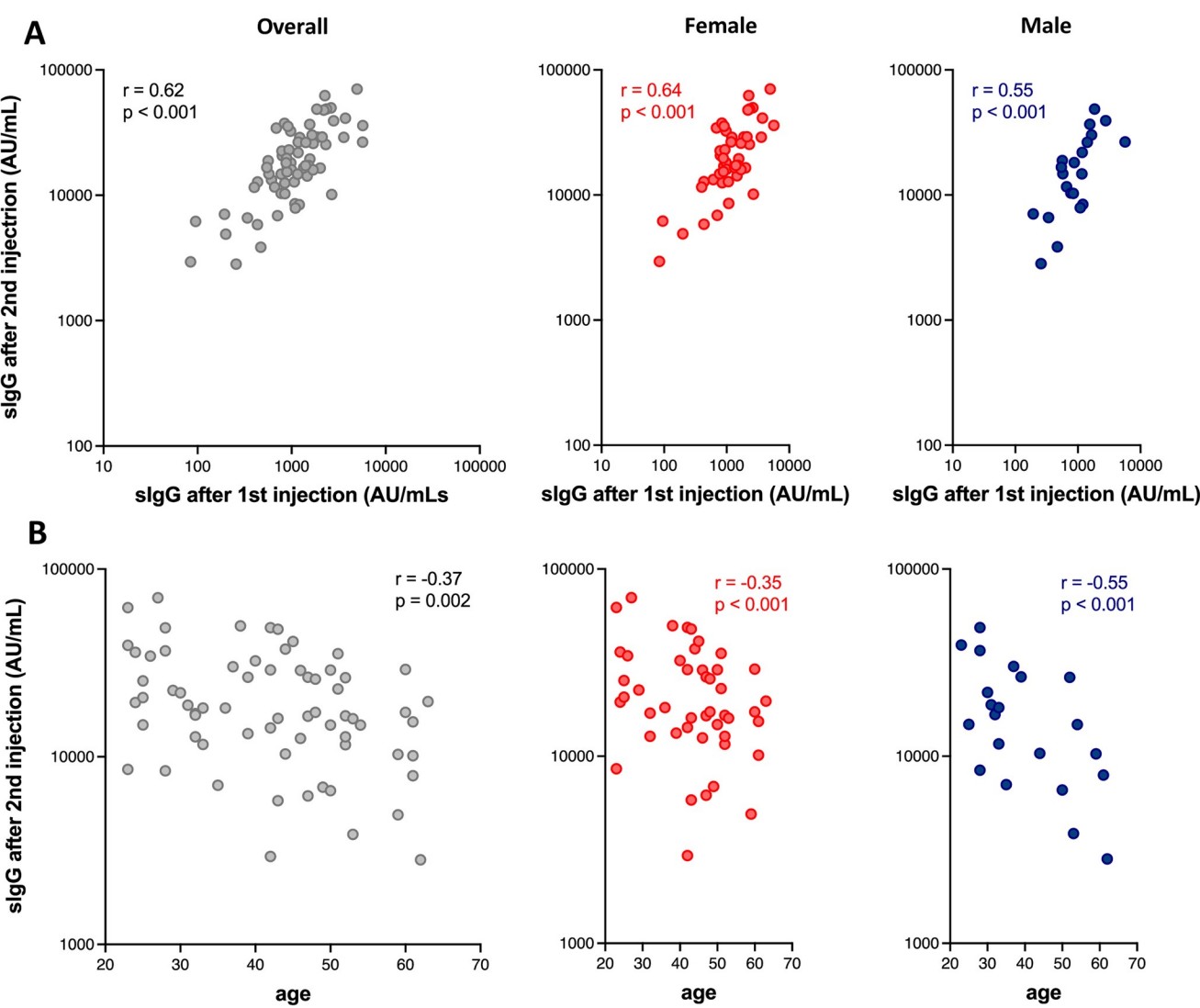

**Fig 2.** A. A linear correlation between spike IgG level after the first injection and that after the second injection in all subjects (left), in female subjects (middle), and in male subjects (right). B. A linear correlation between age and spike IgG level after the second injection in all subjects (left), in female subjects (middle), and in male subjects (right). sIgG, spike IgG.

**Table 2. Univariate linear regression analysis for the association of log-transformed spike IgG two weeks after the second injection of BNT162b2 mRNA vaccine.**

| variables | Beta coefficients (95% CI) | SE | t-value | p-value |
|---|---|---|---|---|
| age | -0.010 (-0.016 –-0.004) | 0.003 | -3.167 | 0.002 |
| male gender | -0.130 (-0.290–0.030) | 0.080 | -1.622 | 0.110 |
| log spike IgG after the first injection | 0.636 (0.497–0.775) | 0.070 | 9.133 | <0.001 |
| AE score after the first injection | 0.025 (-0.026–0.076) | 0.026 | 0.967 | 0.337 |
| AE score after the second injection | 0.057 (0.030–0.085) | 0.014 | 4.119 | <0.001 |

AE, adverse effect; CI, confidence interval; SE, standard error.

**Table 3. Multivariate linear regression analysis of the association of log-transformed spike IgG two weeks after the second injection of BNT162b2 mRNA vaccine.**

| variables | Beta coefficients (95% CI) | SE | t-value | p-value |
|---|---|---|---|---|
| age | -0.006 (-0.010 – -0.001) | 0.002 | -2.640 | 0.011 |
| male gender | -0.093 (-0.199–0.012) | 0.053 | -1.772 | 0.081 |
| log spike IgG after the first injection | 0.550 (0.412–0.688) | 0.069 | 7.977 | <0.001 |
| AE score after the first injection | 0.001 (-0.035–0.036) | 0.018 | 0.031 | 0.976 |
| AE score after the second injection | 0.017 (-0.006–0.041) | 0.012 | 1.491 | 0.141 |

Residual standard error: 0.1879.

Adjusted $R^2$: 0.6285.

F-statistic: 23.33, p<0.001.

BNT162b2 mRNA vaccine (30 μg), which was determined for western subjects, would result in a higher dose for people with smaller body mass, such as Japanese.

The geometric mean of spike IgG after the second injection was 17,378 AU/mL, which was higher than corresponding values measured in a naïve Spanish COVID-19 population [12], but similar to another study [13]. In addition, the range of values was quite broad, from 2,826 AU/mL to 70,272 AU/mL. Although there is no definite cut-off of value for spike IgG sufficient to prevent COVID-19 infection, the FDA authorized only the use of high-titer COVID-19 convalescent plasma (spike IgG > 840 AU/mL), for treatment of patients hospitalized with COVID-19 infection, early in the course of the pandemic [14]. Prendecki et al. reported that one COVID-19 naïve individual whose spike IgG titer was 61.8 AU/mL, developed symptomatic COVID-19 infection 5 weeks after the first dose of BNT162b2 vaccine [15]. The authors also stated that spike IgG titers < 250 AU/mL might not be sufficient for virus neutralization. The lowest IgG titers observed in this study exceeded the latter value by nearly 10-fold. Spike IgG levels after the second injection were higher in subjects < 50 years than in those ≥ 50 years. Previous studies did not show age-dependent efficacy of BNT162b2 for subsequent COVID-19 infection [3, 6]. Thus, these results raise concerns regarding the recommended dose of BNT162b2 mRNA vaccine in younger Japanese people. The current infection situation in Japan makes dose response trials with BNT162b2 vaccine almost impossible. However, if the level of spike IgG three weeks after the first vaccination is > 1,000 AU/mL, the second injection dose could be reduced in younger people, which would enhance the Japanese vaccination rate.

Although efficacy and adverse effects of vaccines appear opposite, they may be related via the strength of the immune response to the vaccine. However, there was no significant association between AE score after the second injection and spike IgG levels after adjusting for age, gender, AE score after the first injection, and spike IgG level after the first injection. Several recent publications and pre-print articles have addressed the relationship between adverse effects and antibody productions. Müller et al. [9] demonstrated that spike IgG and neutralizing antibody titers were significantly lower in patients with > 80 years old than in patients with < 60 years old. Although the prevalence of adverse effects after vaccination was lower in older persons (>80 years old) compared with younger recipients (< 60 years old), there was no correlation between spike IgG levels and post-vaccination adverse effects. Coggins and colleagues [16] calculated AE score and measured spike IgG titers in 206 healthcare workers. AE score was correlated with age, sex, and body weight. However, spike IgG titers were not correlated with AE score after the first injection or the second injection. Debes et al. [17] longitudinally collected spike IgG antibody levels and symptoms after vaccination with either the Moderna or Pfizer/BioNTech vaccines in 954 healthcare workers, and found that clinically

significant symptoms (fever, chills, fatigue) were independently associated with higher median IgG measurements after adjusting for elapsed time after the second vaccination. However, the ratio of IgG antibody measurements was not remarkably different between the two groups (8.82 in subjects with significant symptoms vs. 8.46 in those with no or mild symptom). The authors concluded that regardless of vaccine reactions, either spike mRNA vaccine provides a robust spike antibody response. These studies concur that adverse effects after vaccination do not necessarily reflect subsequent spike IgG production, and more importantly, lack of an adverse effect after the vaccination does not mean that the vaccine did not work effectively [16].

Although the relatively small sample size could produce false negative results, adverse effects after the second injection may be dose-dependent immune reactions after repeated exposure to excipients to stabilize lipid nanoparticles, such as PEG2000, which serves as the vehicle for the vaccine mRNA [8]. Age was negatively correlated with spike IgG production, which agrees with previous studies [3, 9]. Spike IgG levels after the first injection were also positively correlated with spike IgG production after the second injection. However, the model including age and spike IgG after the first vaccination was not sufficiently robust to predict the amount of spike IgG after the second injection with confidence.

## Study limitations

Small sample size limited the generalizability of our results. Since our results were derived from healthy Japanese healthcare workers, further studies should examine subjects with comorbidities and more aged populations [18, 19]. A longitudinal study to determine spike IgG levels after vaccination is another important topic for ongoing research.

## Conclusions

Reactogenicity after the second injection of BNT162b2 mRNA vaccine was common, especially in young to middle aged Japanese healthcare workers. However, its intensity was not closely correlated with subsequent spike IgG levels, after adjusting for age, sex, reactogenicity after the first injection, and spike IgG levels after the first injection. Thus, reactogenicity after the second injection is probably not a reliable measure of antibody production.

## Supporting information

**S1 File. Data file.**
(XLSX)

## Author Contributions

**Conceptualization:** Masaaki Takeuchi.

**Data curation:** Masaaki Takeuchi, Yukie Higa, Akina Esaki, Yosuke Nabeshima, Akemi Nakazono.

**Formal analysis:** Masaaki Takeuchi, Yosuke Nabeshima.

**Funding acquisition:** Masaaki Takeuchi.

**Investigation:** Yukie Higa, Akina Esaki, Akemi Nakazono.

**Methodology:** Masaaki Takeuchi.

**Supervision:** Masaaki Takeuchi.

**Writing – original draft:** Masaaki Takeuchi.

**Writing – review & editing:** Masaaki Takeuchi, Yukie Higa, Akina Esaki, Yosuke Nabeshima, Akemi Nakazono.

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
