## [Decision Letter · Decision Letter 0]

12 Aug 2021

PONE-D-21-19170

Does reactogenicity after a second injection of the BNT162b2 vaccine predict spike IgG antibody levels in healthy Japanese subjects?

PLOS ONE

Dear Dr. Takeuchi,

Thank you for submitting your manuscript to PLOS ONE. After careful consideration, we feel that it has merit but does not fully meet PLOS ONE’s publication criteria as it currently stands. Therefore, we invite you to submit a revised version of the manuscript that addresses the points raised during the review process.

The reviewers have both raised minor points for your consideration. Please address these in your revised version.

We look forward to receiving your revised manuscript.

Kind regards,

Nicole J. Moreland

Academic Editor

PLOS ONE

Journal Requirements:

No.

MT received a research grant from Abbott. Other authors declare that they have no conflicts of interest. 

Additional Editor Comments:

Reviewers' comments:

Reviewer's Responses to Questions

**Comments to the Author**

1. Is the manuscript technically sound, and do the data support the conclusions?

Reviewer #1: Yes

Reviewer #2: Yes

2. Has the statistical analysis been performed appropriately and rigorously? 

Reviewer #1: Yes

Reviewer #2: I Don't Know

3. Have the authors made all data underlying the findings in their manuscript fully available?

Reviewer #1: Yes

Reviewer #2: Yes

4. Is the manuscript presented in an intelligible fashion and written in standard English?

Reviewer #1: Yes

Reviewer #2: Yes

5. Review Comments to the Author

Reviewer #1: Review of “Does reactogenicity after a second injection of the BNT162b2 vaccine predict spike IgG antibody levels in healthy Japanese subjects, Manuscript PONE-D_21-19170"

Page 3, Line 46 “produces a substantial antibody titers” should be either produces substantial antibody titres or produces a substantial antibody titer.

Page 5, Lines 89-94 It took me a longish time to understand the scoring system and then not entirely sure I’ve got it right. I’m assuming for each symptom present there was a score of one; with different measured levels of fever being the exception. I wonder if it’s worth adding a couple of examples of symptom combinations and the resulting score? Alternatively suggest the following wording

Instead of “When local and systemic reactogenicity, except for fever, was present, we scored it as 1.”

Maybe- For each symptom present, whether local or systemic, we assigned a score of 1. The exception was fever...

Reviewer #2: The authors have conducted a small but nonetheless interesting trial of adverse reactions following Pfizer/BioNTech’s BNT162b2 vaccination and asked whether an increase in adverse reactions could relate to an increased reactogenictiy and thus higher antibody response.

Most of the findings of the paper are in agreement with previous studies and the results are interesting. Although as the authors point out, the low numbers included in the trial would need to be validated in larger trials. I note that there are a couple of pre-prints showing the same findings in larger trials such as:

Adverse effects and antibody titers in response to the BNT162b2 mRNA COVID-19 vaccine in a prospective study of healthcare workers, in MedRxiv.

This could be cited in the papers.

The one comment I have is to the statement:

Given the small number of participants, we did not perform formal statistical comparisons between groups.

I would suggest the authors do at least present statistical comparisons and report p-values for correlations, as I suspect the negative correlation between age and IgG levels after second vaccine would not be statically significant, which would need to be discussed further. This observation is backed up by other studies but may not be significant here due to smaller sample size.

6. PLOS authors have the option to publish the peer review history of their article (what does this mean?). If published, this will include your full peer review and any attached files.

Reviewer #1: **Yes: **Dr Janine Paynter

Reviewer #2: No

---

## [Author Response · Author response to Decision Letter 0]

22 Aug 2021

Dear Editors and Reviewers,

We thank all of you for your thoughtful and constructive comments regarding our manuscript entitled,” Does reactogenicity after a second injection of the BNT162b2 vaccine predict spike IgG antibody levels in healthy Japanese subjects? (PONE-D-21-19170)”. Below, in blue, we have provided a point-by-point response to your comments. Red denotes new or revised text in the manuscript

Responses to the Editors:

RESPONSE: We checked that references are complete and correct. Since four pre-prints (medRxiv) cited in the previous manuscript have now been published (Müller, Mazzoni, Velasco, and Narasimham), we replaced the following references:

9. Müller L, Andrée M, Moskorz W, Drexler I, Walotka L, Grothmann R, et al. Age-dependent Immune Response to the Biontech/Pfizer BNT162b2 Coronavirus Disease 2019 Vaccination. Clin Infect Dis. 2021:ciab381. doi: 10.1093/cid/ciab381. PubMed PMID: 33906236; PubMed Central PMCID: PMC8135422.

10. Mazzoni A, Di Lauria N, Maggi L, Salvati L, Vanni A, Capone M, et al. First dose mRNA vaccination is sufficient to reactivate immunological memory to SARS-CoV-2 in subjects who have recovered from COVID-19. J Clin Invest. 2021;131(12):e149150. doi: 10.1172/JCI149150. PubMed PMID: 33939647; PubMed Central PMCID: PMC8203460.

12. Velasco M, Galán MI, Casas ML, Pérez-Fernández E, Martínez-Ponce D, González-Piñeiro B, et al. Impact of previous coronavirus disease 2019 on immune response after a single dose of BNT162b2 severe acute respiratory syndrom coronavirus 2 vaccine. Open Forum Infect Dis. 2021;8(7). doi: 10.1093/ofid/ofab299. PubMed PMID: 34258322; PubMed Central PMCID: PMC8244747.

13. Narasimhan M, Mahimainathan L, Araj E, Clark AE, Markantonis J, Green A, et al. Clinical Evaluation of the Abbott Alinity SARS-CoV-2 Spike-Specific Quantitative IgG and IgM Assays among Infected, Recovered, and Vaccinated Groups. J Clin Microbiol. 2021;59(7):e0038821. doi: 10.1128/JCM.00388-21. PubMed PMID: 33827901; PubMed Central PMCID: PMC8218760.

We added two references in response to the comments from reviewer #2.

16. Coggins SAA, Laing ED, Olsen CH, Goguet E, Moser M, Jackson-Thompson BM, et al. Adverse effects and antibody titers in response to the BNT162b2 mRNA COVID-19 vaccine in a prospective study of healthcare workers. medRxiv [Preprint]. 2021. doi: https://doi.org/10.1101/2021.06.25.21259544.

17. Debes AK, Xiao S, Colantuoni E, Egbert ER, Caturegli P, Gadala A, et al. Association of Vaccine Type and Prior SARS-CoV-2 Infection With Symptoms and Antibody Measurements Following Vaccination Among Health Care Workers. JAMA Intern Med. 2021. doi: 10.1001/jamainternmed.2021.4580. PubMed PMID: 34398173.

Response to the reviewer #1:

Page 3, Line 46 “produces a substantial antibody titers” should be either produces substantial antibody titers or produces a substantial antibody titer.

RESPONSE: Thank you for pointing this out. We changed this to “produces substantial antibody titers”.

Page 3, Lines 46-47:

The second injection has a booster effect that produces substantial antibody titers against SARS-CoV2 spike antigen.

Page 5, Line 89-94 It took me a longish time to understand the scoring system and then not entirely sure I’ve got it right. I’m assuming for each symptom present there was a score of one; with different measured levels of fever being the exception. I wonder if it’s worth adding a couple of examples of symptom combinations and the resulting score? Alternatively suggested the following wording

Instead of “When local and systemic reactogenicity, except for fever, was present, we score it as 1.”

Maybe- For each symptom present, whether local and systemic, we assigned a score of 1. The exception was fever…

RESPONSE: Thank you for the thoughtful comments. We agree that the weight of each symptom may be different. However, we also don’t know which symptom is more important. Thus, we arbitrarily scored for adverse effects. A similar scoring system has been used in the recent publications (ref. #16), but definitions vary among studies (ref. #16, #17). We corrected the sentences as suggested.

Page 5, Lines 93-94

For each symptom present, whether local and systemic, we assigned a score of 1. The exception was fever. A score  

Responses to the Reviewer #2:

The authors have conducted a small but nonetheless interesting trial of adverse reactions following Pfizer/BioNTech’s BNT162b2 vaccination and asked whether an increase in adverse reactions could relate to an increased reactogenicity and thus higher antibody response.

Most of the findings of the paper are in agreement with previous studies and the results are interesting. Although as the authors pointed out, the low numbers included in the trial would need to be validated in larger trials. I note that there are a couple of pre-prints showing the same findings in larger trials such as:

Adverse effects and antibody titers in response to the BNT162b2 mRNA COVID-19 vaccine in a prospective study of healthcare workers, in MedRxiv. This could be cited in the papers.

RESPONSE: Thank you for the positive comments and important suggestions. We incorporated recent publications or pre-prints in this revised manuscript and discussed reactogenicity and antibody production.

Page 13-14, Discussion, Lines 239-259:

Several recent publications and pre-print articles have addressed the relationship between adverse effects and antibody productions. Müller et al. [9] demonstrated that spike IgG and neutralizing antibody titers were significantly lower in patients with > 80 years old than in patients with < 60 years old. Although the prevalence of adverse effects after vaccination was lower in older persons (>80 years old) compared with younger recipients (< 60 years old), there was no correlation between spike IgG levels and post-vaccination adverse effects. Coggins and colleagues [16] calculated AE score and measured spike IgG titers in 206 healthcare workers. AE score was correlated with age, sex, and body weight. However, spike IgG titers were not correlated with AE score after the first injection or the second injection. Debes et al. [17] longitudinally collected spike IgG antibody levels and symptoms after vaccination with either the Moderna or Pfizer/BioNTech vaccines in 954 healthcare workers, and found that clinically significant symptoms (fever, chills, fatigue) were independently associated with higher median IgG measurements after adjusting for elapsed time after the second vaccination. However, the ratio of IgG antibody measurements was not remarkably different between the two groups (8.82 in subjects with significant symptoms vs. 8.46 in those with no or mild symptom). The authors concluded that regardless of vaccine reactions, either spike mRNA vaccine provides a robust spike antibody response. These studies concur that adverse effects after vaccination do not necessarily reflect subsequent spike IgG production, and more importantly, lack of an adverse effect after the vaccination does not mean that the vaccine did not work effectively [16].

Page 17, References, Lines 344-351:

16. Coggins SAA, Laing ED, Olsen CH, Goguet E, Moser M, Jackson-Thompson BM, et al. Adverse effects and antibody titers in response to the BNT162b2 mRNA COVID-19 vaccine in a prospective study of healthcare workers. medRxiv [Preprint]. 2021. doi: https://doi.org/10.1101/2021.06.25.21259544.

17. Debes AK, Xiao S, Colantuoni E, Egbert ER, Caturegli P, Gadala A, et al. Association of Vaccine Type and Prior SARS-CoV-2 Infection With Symptoms and Antibody Measurements Following Vaccination Among Health Care Workers. JAMA Intern Med. 2021. doi: 10.1001/jamainternmed.2021.4580. PubMed PMID: 34398173.

The one comment I have is to the statement:

Given the small number of participants, we did not perform formal statistical comparisons between groups.

I would suggest the authors do at least present statistical comparisons and report p-values for correlations, as I suspect the negative correlation between age and IgG levels after second vaccine would not be statistically significant, which would need to be discussed further. This observation is backed up by the other studies but may not be significant here due to smaller sample size.

RESPONSE: Thank you for the thoughtful comments. The reason why we did not perform statistical comparisons between groups (Table 1) is a problem of “multiplicity”. Our primary endpoint was to determine whether the severity of adverse effects after the vaccination predicts the amount of antibody production. Thus, we only performed statistical analysis for this purpose. However, we agreed with your comments and incorporated p-values for correlations in this revised manuscript. Although the number of subjects was small, we found that the correlation between age and spike IgG after the second vaccination was statistically significant (p=0.002). We incorporated the results in the manuscript text, and in Figure 2.

Page 9, Lines 154-157:

There was a positive correlation between spike IgG levels after the first injection and after the second injection (r = 0.62, p < 0.001, Fig 2A). There was also a negative correlation between age and spike IgG levels after the second injection (r = -0.37, p = 0.002, Fig 2B).

---

## [Decision Letter · Decision Letter 1]

8 Sep 2021

Does reactogenicity after a second injection of the BNT162b2 vaccine predict spike IgG antibody levels in healthy Japanese subjects?

PONE-D-21-19170R1

Dear Dr. Takeuchi,

We’re pleased to inform you that your manuscript has been judged scientifically suitable for publication and will be formally accepted for publication once it meets all outstanding technical requirements.

Kind regards,

Nicole J. Moreland

Academic Editor

PLOS ONE

Additional Editor Comments (optional):

All comments have been addressed in the revised version and I am happy to accept this paper for publication.

Reviewers' comments:

Reviewer's Responses to Questions

**Comments to the Author**

1. If the authors have adequately addressed your comments raised in a previous round of review and you feel that this manuscript is now acceptable for publication, you may indicate that here to bypass the “Comments to the Author” section, enter your conflict of interest statement in the “Confidential to Editor” section, and submit your "Accept" recommendation.

Reviewer #2: All comments have been addressed

2. Is the manuscript technically sound, and do the data support the conclusions?

Reviewer #2: Yes

3. Has the statistical analysis been performed appropriately and rigorously? 

Reviewer #2: Yes

4. Have the authors made all data underlying the findings in their manuscript fully available?

Reviewer #2: Yes

5. Is the manuscript presented in an intelligible fashion and written in standard English?

Reviewer #2: Yes

6. Review Comments to the Author

Reviewer #2: (No Response)

7. PLOS authors have the option to publish the peer review history of their article (what does this mean?). If published, this will include your full peer review and any attached files.

Reviewer #2: No

---

## [Editor Report · Acceptance letter]

10 Sep 2021

PONE-D-21-19170R1 

Does reactogenicity after a second injection of the BNT162b2 vaccine predict spike IgG antibody levels in healthy Japanese subjects? 

Dear Dr. Takeuchi:

I'm pleased to inform you that your manuscript has been deemed suitable for publication in PLOS ONE. Congratulations! Your manuscript is now with our production department. 

Kind regards, 

on behalf of

Dr. Nicole J. Moreland 

Academic Editor

PLOS ONE